# On the Consistency of Quick Shift

**Heinrich Jiang**
Google Inc.
1600 Amphitheatre Parkway, Mountain View, CA 94043
`heinrich.jiang@gmail.com`

## Abstract

Quick Shift is a popular mode-seeking and clustering algorithm. We present finite sample statistical consistency guarantees for Quick Shift on mode and cluster recovery under mild distributional assumptions. We then apply our results to construct a consistent modal regression algorithm.

## 1 Introduction

Quick Shift [16] is a clustering and mode-seeking procedure that has received much attention in computer vision and related areas. It is simple and proceeds as follows: it moves each sample to its closest sample with a higher empirical density if one exists in a $\tau$ radius ball, where the empirical density is taken to be the Kernel Density Estimator (KDE). The output of the procedure can thus be seen as a graph whose vertices are the sample points and a directed edge from each sample to its next point if one exists. Furthermore, it can be seen that Quick Shift partitions the samples into trees which can be taken as the final clusters, and the root of each such tree is an estimate of a local maxima.

Quick Shift was designed as an alternative to the better known mean-shift procedure [4, 5]. Mean-shift performs a gradient ascent of the KDE starting at each sample until $\epsilon$-convergence. The samples that correspond to the same points of convergence are in the same cluster and the points of convergence are taken to be the estimates of the modes. Both procedures aim at clustering the data points by incrementally hill-climbing to a mode in the underlying density. Some key differences are that Quick Shift restricts the steps to sample points and has the extra $\tau$ parameter. In this paper, we show that Quick Shift can surprisingly attain strong statistical guarantees without the second-order density assumptions required to analyze mean-shift.

We prove that Quick Shift recovers the modes of an arbitrary multimodal density at a minimax optimal rate under mild nonparametric assumptions. This provides an alternative to known procedures with similar statistical guarantees; however such procedures only recover the modes but fail to inform us how to assign the sample points to a mode which is critical for clustering. Quick Shift on the other hand recovers both the modes and the clustering assignments with statistical consistency guarantees. Moreover, Quick Shift's ability to do all of this has been extensively validated in practice.

A unique feature of Quick Shift is that it has a segmentation parameter $\tau$ which allows practioners to merge clusters corresponding to certain less salient modes of the distribution. In other words, if a local mode is not the maximizer of its $\tau$-radius neighborhood, then its corresponding cluster will become merged to that of another mode. Current consistent mode-seeking procedures [6, 12] fail to allow one to control such segmentation. We give guarantees on how Quick Shift does this given an arbitrary setting of $\tau$.

We show that Quick Shift can also be used to recover the cluster tree. In cluster tree estimation, the known procedures with the strongest statistical consistency guarantees include Robust Single Linkage (RSL) [2] and its variants e.g. [13, 7]. We show that Quick Shift attains similar guarantees.

Thus, Quick Shift, a simple and already popular procedure, can *simultaneously* recover the modes with segmentation tuning, provide clustering assignments to the appropriate mode, and can estimate the cluster tree of an unknown density $f$ with the strong consistency guarantees. No other procedure has been shown to have these properties.

Then we use Quick Shift to solve the modal regression problem [3], which involves estimating the *modes* of the conditional density $f(y|X)$ rather than the mean as in classical regression. Traditional approaches use a modified version of mean-shift. We provide an alternative using Quick Shift which has precise statistical consistency guarantees under much more mild assumptions.

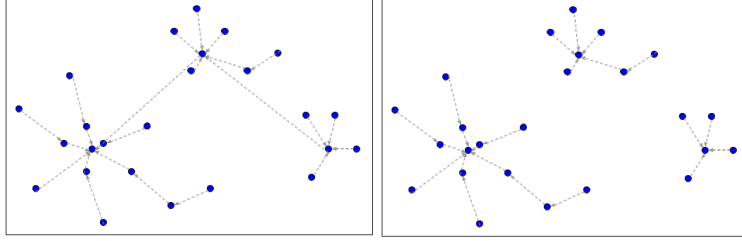

Figure 1: Quick Shift example. **Left**: $\tau = \infty$. The procedure returns one tree, whose head is the sample with highest empirical density. **Right**: $\tau$ set to a lower value. The edges with length greater than $\tau$ are no longer present when compared to the left. We are left with three clusters.

## 2  Assumptions and Supporting Results

---
**Algorithm 1** Quick Shift
---
Input: Samples $X_{[n]} := \{x_1, ..., x_n\}$, KDE bandwidth $h$, segmentation parameter $\tau > 0$.
Initialize directed graph $G$ with vertices $\{x_1, ..., x_n\}$ and no edges.
**for** $i = 1$ to $n$ **do**
  **if** there exists $x \in X_{[n]}$ such that $\widehat{f}_h(x) > \widehat{f}_h(x_i)$ and $||x - x_i|| \leq \tau$ **then**
    Add to $G$ a directed edge from $x_i$ to $\operatorname{argmin}_{x_j \in X_{[n]}: \widehat{f}_h(x_j) > \widehat{f}_h(x_i)} ||x_i - x_j||$.
  **end if**
**end for**
**return** $G$.

---

### 2.1  Setup

Let $X_{[n]} = \{x_1, ..., x_n\}$ be $n$ i.i.d. samples drawn from distribution $\mathcal{F}$ with density $f$ over the uniform measure on $\mathbb{R}^d$.

**Assumption 1** (Hölder Density). *$f$ is Hölder continuous on compact support $\mathcal{X} \subseteq \mathbb{R}^d$. i.e. $|f(x) - f(x')| \leq C_\alpha ||x - x'||^\alpha$ for all $x, x' \in \mathcal{X}$ and some $0 < \alpha \leq 1$ and $C_\alpha > 0$.*

**Definition 1** (Level Set). *The $\lambda$ level set of $f$ is defined as $L_f(\lambda) := \{x \in \mathcal{X} : f(x) \geq \lambda\}$.*

**Definition 2** (Hausdorff Distance). *$d_H(A, A') = \max\{\sup_{x \in A} d(x, A'), \sup_{x \in A'} d(x, A)\}$, where $d(x, A) := \inf_{x' \in A} ||x - x'||$.*

The next assumption says that the level sets are continuous w.r.t. the level in the following sense where we denote the $\epsilon$-interior of $A$ as $A^{\ominus \epsilon} := \{x \in A, \inf_{y \in \partial A} d(x, y) \geq \epsilon\}$ ($\partial A$ is the boundary of $A$):

**Assumption 2** (Uniform Continuity of Level Sets). *For each $\epsilon > 0$, there exists $\delta > 0$ such that for $0 < \lambda \leq \lambda' \leq ||f||_\infty$ with $|\lambda - \lambda'| < \delta$, then $L_f(\lambda)^{\ominus \epsilon} \subseteq L_f(\lambda')$.*

**Remark 1.** *Procedures that try to incrementally move points to nearby areas of higher density will have difficulties in regions where there is little or no change in density. The above assumption is a simple and mild formulation which ensures there are no such flat regions.*

**Remark 2.** *Note that our assumptions are quite mild when compared to analyses of similar procedures like mean-shift, which require at least second-order smoothness assumptions. Interestingly, we only require Hölder continuity.*

## 2.2 KDE Bounds

We next give uniform bounds on KDE required to analyze Quick Shift.

**Definition 3.** *Define kernel function* $K : \mathbb{R}^d \to \mathbb{R}_{\geq 0}$ *where* $\mathbb{R}_{\geq 0}$ *denotes the non-negative real numbers such that* $\int_{\mathbb{R}^d} K(u) du = 1$.

We make the following mild regularity assumptions on $K$.

**Assumption 3.** *(Spherically symmetric, non-increasing, and exponential decays) There exists non-increasing function* $k : \mathbb{R}_{\geq 0} \to \mathbb{R}_{\geq 0}$ *such that* $K(u) = k(|u|)$ *for* $u \in \mathbb{R}^d$ *and there exists* $\rho, C_\rho, t_0 > 0$ *such that for* $t > t_0$, $k(t) \leq C_\rho \cdot \exp(-t^\rho)$.

**Remark 3.** *These assumptions allow the popular kernels such as Gaussian, Exponential, Silverman, uniform, triangular, tricube, Cosine, and Epanechnikov.*

**Definition 4** (Kernel Density Estimator). *Given a kernel* $K$ *and bandwidth* $h > 0$ *the KDE is defined by*

$$\widehat{f}_h(x) = \frac{1}{n \cdot h^d} \sum_{i=1}^n K\left(\frac{x - X_i}{h}\right).$$

Here we provide the uniform KDE bound which will be used for our analysis, established in [11].

**Lemma 1.** *[$\ell_\infty$ bound for $\alpha$-Hölder continuous functions. Theorem 2 of [11]] There exists positive constant* $C'$ *depending on* $f$ *and* $K$ *such that the following holds with probability at least* $1 - 1/n$ *uniformly in* $h > (\log n / n)^{1/d}$.

$$\sup_{x \in \mathbb{R}^d} |\widehat{f}_h(x) - f(x)| < C' \cdot \left( h^\alpha + \sqrt{\frac{\log n}{n \cdot h^d}} \right).$$

# 3 Mode Estimation

In this section, we give guarantees about the local modes returned by Quick Shift. We make the additional assumption that the modes are local maxima points with negative-definite Hessian.

**Assumption 4.** *[Modes] A local maxima of* $f$ *is a connected region* $M$ *such that the density is constant on* $M$ *and decays around its boundaries. Assume that each local maxima of* $f$ *is a point, which we call a mode. Let* $\mathcal{M}$ *be the modes of* $f$ *where* $\mathcal{M}$ *is a finite set. Then let* $f$ *be twice differentiable around a neighborhood of each* $x \in \mathcal{M}$ *and let* $f$ *have a negative-definite Hessian at each* $x \in \mathcal{M}$ *and those neighborhoods are disjoint.*

This assumption leads to the following.

**Lemma 2** (Lemma 5 of [6]). *Let* $f$ *satisfy Assumption 4. There exists* $r_M, \check{C}, \hat{C} > 0$ *such that the following holds for all* $x_0 \in \mathcal{M}$ *simultaneously.*

$$\check{C} \cdot |x_0 - x|^2 \leq f(x_0) - f(x) \leq \hat{C} \cdot |x_0 - x|^2,$$

*for all* $x \in A_{x_0}$ *where* $A_{x_0}$ *is a connected component of* $\{x : f(x) \geq \inf_{x' \in B(x_0, r_M)} f(x)\}$ *which contains* $x_0$ *and does not intersect with other modes.*

The next assumption ensures that the level sets don't become arbitrarily thin as long as we are sufficiently away from the modes.

**Assumption 5.** *[Level Set Regularity] For each* $\sigma, r > 0$, *there exists* $\eta > 0$ *such that the following holds for all connected components* $A$ *of* $L_f(\lambda)$ *with* $\lambda > 0$ *and* $A \not\subseteq \cup_{x_0 \in \mathcal{M}} B(x_0, r)$. *If* $x$ *lies on the boundary of* $A$, *then* $Vol(B(x, \sigma) \cap A) > \eta$ *where Vol is volume w.r.t. the uniform measure on* $\mathbb{R}^d$.

We next give the result about mode recovery for Quick Shift. It says that as long as $\tau$ is small enough, then as the number of samples grows, the roots of the trees returned by Quick Shift will bijectively correspond to the true modes of $f$ and the estimation errors will match lower bounds established by Tsybakov [15] up to logarithmic factors. We defer the proof to Theorem 2 which is a generalization of the following result.

**Theorem 1** (Mode Estimation guarantees for Quick Shift). *Let* $\tau < r_M/2$ *and Assumptions 1, 2, 3, 4, and 5 hold. Choose* $h$ *such that* $(\log n)^{2/\rho} \cdot h \to 0$ *and* $\log n / (nh^d) \to 0$ *as* $n \to \infty$. *Let* $\widehat{\mathcal{M}}$ *be the*

*heads of the trees in $G$ (returned by Algorithm 1). There exists constant $C$ depending on $f$ and $K$ such that for $n$ sufficiently large, with probability at least $1 - 1/n$,*

$$d_H(\mathcal{M}, \widehat{\mathcal{M}})^2 < C\left((\log n)^{4/\rho}h^2 + \sqrt{\frac{\log n}{n \cdot h^d}}\right).$$

*and $|\mathcal{M}| = |\widehat{\mathcal{M}}|$. In particular, taking $h \approx n^{-1/(4+d)}$ optimizes the above rate to $d(\mathcal{M}, \widehat{\mathcal{M}}) = \tilde{O}(n^{-1/(4+d)})$. This matches the minimax optimal rate for mode estimation up to logarithmic factors.*

We now give a stronger notion of mode that fits better for analyzing the role of $\tau$. In the last result, it was assumed that the practitioner wished to recover exactly the modes of the density $f$ by taking $\tau$ sufficiently small. Now, we analyze the case where $\tau$ is intentionally set to a particular value so that Quick Shift produces segmentations that group modes together that are in close proximity to higher density regions.

**Definition 5.** *A mode $x_0 \in \mathcal{M}$ is an $(r,\delta)^+$-mode if $f(x_0) > f(x) + \delta$ for all $x \in B(x_0, r)\backslash B(x_0, r_M)$. A mode $x_0 \in \mathcal{M}$ is an $(r,\delta)^-$-mode if $f(x_0) < f(x) - \delta$ for some $x \in B(x_0, r)$. Let $\mathcal{M}^+_{r,\delta} \subseteq \mathcal{M}$ and $\mathcal{M}^-_{r,\delta} \subseteq \mathcal{M}$ denote the set of $(r,\delta)^+$-modes and $(r,\delta)^-$-modes of $f$, respectively.*

In other words, an $(r,\delta)^+$-mode is a mode that is also a maximizer in a larger ball of radius $r$ by at least $\delta$ when outside of the region where there is quadratic decay and smoothness ($B(x_0, r_M)$). An $(r,\delta)^-$-mode is a mode that is not the maximizer in its radius $r$ ball by a margin of at least $\delta$.

The next result shows that Algorithm recovers the $(\tau + \epsilon, \delta)^+$-modes of $f$ and excludes the $(\tau - \epsilon, \delta)^-$-modes of $f$. The proof is in the appendix.

**Theorem 2.** *(Generalization of Theorem 1) Let $\delta, \epsilon > 0$ and suppose Assumptions 1, 2, 3, 4, and 5 hold. Let $h \equiv h(n)$ be chosen such that $h \to 0$ and $\log n/(nh^d) \to 0$ as $n \to \infty$. Then there exists $C > 0$ depending on $f$ and $K$ such that the following holds for $n$ sufficiently large with probability at least $1 - 1/n$. For each $x_0 \in \mathcal{M}^+_{\tau+\epsilon,\delta}\backslash\mathcal{M}^-_{\tau-\epsilon,\delta}$, there exists unique $\hat{x} \in \widehat{\mathcal{M}}$ such that*

$$||x_0 - \hat{x}||^2 < C\left((\log n)^{4/\rho}h^2 + \sqrt{\frac{\log n}{n \cdot h^d}}\right).$$

*Moreover, $|\widehat{\mathcal{M}}| \le |\mathcal{M}| - |\mathcal{M}^-_{\tau-\epsilon,\delta}|$.*
*In particular, taking $\epsilon \to 0$ and $\delta \to 0$ gives us an exact characterization of the asymptotic behavior of Quick Shift in terms of mode recovery.*

## 4 Assignment of Points to Modes

In this section, we give guarantees on how the points are assigned to their respective modes. We first give the following definition which formalizes how two points are separated by a wide and deep valley.

**Definition 6.** *$x_1, x_2 \in \mathcal{X}$ are $(r_s, \delta)$-separated if there exists a set $S$ such that every path from $x_1$ and $x_2$ intersects with $S$ and*

$$\sup_{x \in S+B(0,r_s)} f(x) < \inf_{x \in B(x_1,r_s) \cup B(x_2,r_s)} f(x) - \delta.$$

**Lemma 3.** *Suppose Assumptions 1, 2, 3, 4, and 5 hold. Let $\tau < r_s/2$ and choose $h$ such that $(\log n)^{2/\rho} \cdot h \to 0$ and $\log n/(nh^d) \to 0$ as $n \to \infty$. Let $G$ be the output of Algorithm 1. The following holds with probability at least $1 - 1/n$ for $n$ sufficiently large depending on $f$, $K$, $\delta$, and $\tau$ uniformly in all $x_1, x_2 \in \mathcal{X}$. If $x_1$ and $x_2$ are $(r_s, \delta)$-separated, then there cannot exist a directed path from $x_1$ to $x_2$ in $G$.*

*Proof.* Suppose that $x_1$ and $x_2$ are $(r_s, \delta)$-separated (with respect to set $S$) and there exists a directed path from $x_1$ to $x_2$ in $G$. Given our choice of $\tau$, there exists some point $x \in G$ such that $x \in S + B(0, r_s)$ and $x$ is on the path from $x_1$ to $x_2$. We have $f(x) < f(x_1) - \delta$. Choose $n$ sufficiently large such that by Lemma 1, $\sup_{x \in \mathcal{X}} |\hat{f}_h(x) - f(x)| < \delta/2$. Thus, we have $\hat{f}_h(x) < \hat{f}_h(x_1)$, which means a directed path in $G$ starting from $x_1$ cannot contain $x$, a contradiction. The result follows. □

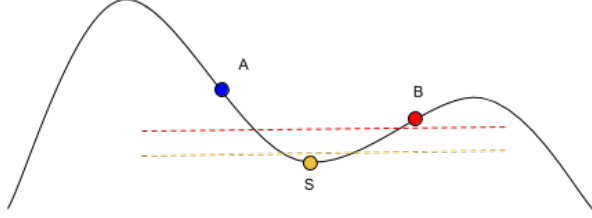

Figure 2: Illustration of $(r_s, \delta)$-separation in 1 dimension. Here $A$ and $B$ are $(r_s, \delta)$-separated by $S$. This is because the minimum density level of $r_s$-radius balls around $A$ and $B$ (the red dotted line) exceeds the maximum density level of the $r_s$-radius ball around $S$ by at least $\delta$ (golden dotted line). In other words, there exists a sufficiently wide (controlled by $r_s$ and $S$) and deep (controlled by $\delta$) valley separating $A$ and $B$. The results in this section will show that in such cases, these pairs of points will not be assigned to the same cluster.

This leads to the following consequence about how samples are assigned to their respective modes.

**Theorem 3.** *Assume the same conditions as Lemma 3. The following holds with probability at least $1 - 1/n$ for $n$ sufficiently large depending on $f$, $K$, $\delta$, and $\tau$ uniformly in $x \in \mathcal{X}$ and $x_0 \in \mathcal{M}$. For each $x \in \mathcal{X}$ and $x_0 \in \mathcal{M}$, if $x$ and $x_0$ are $(r_s, \delta)$-separated, then $x$ will not be assigned to the tree corresponding to $x_0$ from Theorem 1.*

**Remark 4.** *In particular, taking $\delta \to 0$ and $r_s \to 0$ gives us guarantees for all points which have a unique mode in which it can be assigned to.*

We now give a more general version of $(r_s, \delta)$-separation, in which the condition holds if every path between the two points dips down at some point. The same results as the above extend for this definition in a straightforward manner.

**Definition 7.** *$x_1, x_2 \in \mathcal{X}$ are $(r_s, \delta)$-weakly-separated if there exists a set $S$, with $x_1, x_2 \notin S + B(0, r_s)$, such that every path $\mathcal{P}$ from $x_1$ and $x_2$ satisifes the following. (1) $\mathcal{P} \cap S \neq \emptyset$ and (2)*

$$\sup_{x \in \mathcal{P} \cap S + B(0, r_s)} f(x) < \inf_{x \in B(x_1', r_s) \cup B(x_2', r_s)} f(x) - \delta,$$

*where $x_1', x_2'$ are defined as follows. Let $\mathcal{P}_1$ be the path obtained by starting at $x_1$ and following $\mathcal{P}$ until it intersects $S$, and $\mathcal{P}_2$ be the path obtained by following $\mathcal{P}$ starting from the last time it intersects $S$ until the end. Then $x_1'$ and $x_2'$ are points which respectively attain the highest values of $f$ on $\mathcal{P}_1$ and $\mathcal{P}_2$.*

## 5 Cluster Tree Recovery

The connected components of the level sets as the density level varies forms a hierarchical structure known as the cluster tree.

**Definition 8** (Cluster Tree). *The cluster tree of $f$ is given by*

$$C_f(\lambda) := \text{connected components of } \{x \in X : f(x) \geq \lambda\}.$$

**Definition 9.** *Let $G(\lambda)$ be the subgraph of $G$ with vertices $x \in X_{[n]}$ such that $\widehat{f}_h(x) > \lambda$ and edges between pairs of vertices which have corresponding edges in $G$. Let $\tilde{G}(\lambda)$ be the sets of vertices corresponding to the connected components of $G(\lambda)$.*

**Definition 10.** *Suppose that $\mathcal{A}$ is a collection of sets of points in $\mathbb{R}^d$. Then define $Link(\mathcal{A}, \delta)$ to be the result of repeatedly removing pairs $A_1, A_2 \in \mathcal{A}$ from $\mathcal{A}$ ($A_1 \neq A_2$) that satisfy $\inf_{a_1 \in A_1} \inf_{a_2 \in A_2} ||a_1 - a_2|| < \delta$ and adding $A_1 \cup A_2$ to $\mathcal{A}$ until no such pairs exist.*

**Parameter settings for Algorithm 2:** Suppose that $\tau \equiv \tau(n)$ is chosen as a function of $n$ such that $\tau \to 0$ as $n \to \infty$, $\tau(n) \geq (\log^2 n/n)^{1/d}$ and $h \equiv h(n)$ is chosen such that $h \to 0$ and $\log n/(nh^d) \to 0$ as $n \to \infty$.

The following is the main result of this section, the proof is in the appendix.

---
**Algorithm 2** Quick Shift Cluster Tree Estimator
---
Input: Samples $X_{[n]} := \{X_1, ..., X_n\}$, KDE bandwidth $h$, segmentation parameter $\tau > 0$.
Let $G$ be the output of Quick Shift (Algorithm 1) with above parameters.
For $\lambda > 0$, let $\widehat{C}_f(\lambda) := \text{Link}(\tilde{G}(\lambda), \tau)$.
**return** $\widehat{C}_f$
---

**Theorem 4** (Consistency). *Algorithm 2 converges in probability to the true cluster tree of $f$ under merge distortion (defined in [7]).*

**Remark 5.** *By combining the result of this section with the mode estimation result, we can obtain the following interpretation. For any level $\lambda$, a component in $G(\lambda)$ estimates a connected component of the $\lambda$-level set of $f$, and that further, the trees within that component in $G(\lambda)$ have a one-to-one correspondence with the modes in the connected component.*

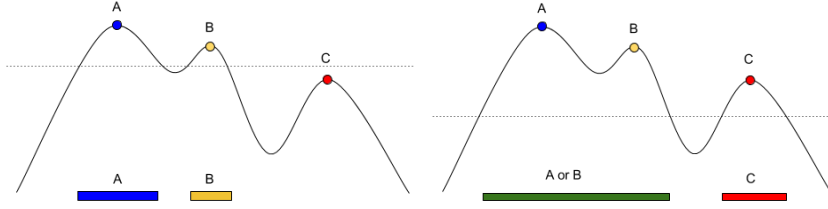

Figure 3: Illustration on 1-dimensional density with three modes $A$, $B$, and $C$. When restricting Quick Shift's output to samples have empirical density above a certain threshold and connecting nearby clusters, then this approximates the connected components of the true density level set. Moreover, we give guarantees that such points will be assigned to clusters which correspond to modes within its connected component.

# 6 Modal Regression

Suppose that we have joint density $f(X, y)$ on $\mathbb{R}^d \times \mathbb{R}$ w.r.t. to the Lebesgue measure. In modal regression, we are interested in estimating the modes of the conditional $f(y|X = x)$ given samples from the joint distribution.

---
**Algorithm 3** Quick Shift Modal Regression
---
Input: Samples $\mathcal{D} := \{(x_1, y_1), ..., (x_n, y_n)\}$, bandwidth $h$, $\tau > 0$, and $x \in \mathcal{X}$.
Let $Y = \{y_1, ..., y_n\}$ and $\widehat{f}_h$ be the KDE computed w.r.t. $\mathcal{D}$.
Initialize directed graph $G$ with vertices $Y$ and no edges.
**for** $i = 1$ to $n$ **do**
    **if** there exists $y_j \in [y_i - \tau, y_i + \tau] \cap Y$ such that $\widehat{f}_h(x, y_j) > \widehat{f}_h(x, y_i)$ **then**
        Add to $G$ an directed edge from $y_i$ to $\text{argmin}_{y_i \in Y: \widehat{f}_h(x, y_j) > \widehat{f}_h(x, y_i)} ||y_i - y_j||$.
    **end if**
**end for**
**return** The roots of the trees of $G$ as the estimates of the modes of $f(y|X = x)$.
---

**Theorem 5** (Consistency of Quick Shift Modal Regression). *Suppose that $\tau \equiv \tau(n)$ is chosen as a function of $n$ such such that $\tau \to 0$ as $n \to \infty$, $\tau(n) \geq (\log^2 n/n)^{1/d}$ and $h \equiv h(n)$ is chosen such that $h \to 0$ and $\log n/(nh^{d+1}) \to 0$ as $n \to \infty$. Let $\mathcal{M}_x$ be the modes of the conditional density $f(y|X = x)$ and $\widehat{\mathcal{M}}_x$ be the output of Algorithm 3. Then with probability at least $1 - 1/n$ uniformly in $x$ such that $f(y|X = x)$ and $K$ satisfies Assumptions 1, 2, 3, 4, and 5,*

$$d_H(\mathcal{M}_x, \widehat{\mathcal{M}}_x) \to 0 \quad as \ n \to \infty.$$

# 7 Related Works

**Mode Estimation.** Perhaps the most popular procedure to estimate the modes is mean-shift; however, it has proven quite difficult to analyze. Arias-Castro et al. [1] made much progress by utilizing dynamical systems theory to show that mean-shift's updates converge to the correct gradient ascent steps. The recent work of Dasgupta and Kpotufe [6] was the first to give a procedure which recovers the modes of a density with minimax optimal statistical guarantees in a multimodal density. They do this by using a top-down traversal of the density levels of a proximity graph, borrowing from work in cluster tree estimation. The procedure was shown to recover exactly the modes of the density at minimax optimal rates.

In this work, we showed that Quick Shift attains the same guarantees while being a simpler approach than known procedures that attain these guarantees [6, 12]. Moreover unlike these procedures, Quick Shift also assigns the remaining samples to their appropriate modes. Furthermore, Quick Shift also has a segmentation tuning parameter $\tau$ which allows us to merge the clusters of modes that are not maximal in its $\tau$-radius neighborhood into the clusters of other modes. This is useful as in practice, one may not wish to pick up every single local maxima, especially when there are local maxima that can be grouped together by proximity. We formalized the segmentation of such modes and identify which modes get returned and which ones become merged into other modes' clusters by Quick Shift.

**Cluster Tree Estimation.** Work on cluster tree estimation has a long history. Some early work on density based clustering by Hartigan [9] modeled the clusters of a density as the regions $\{x : f(x) \geq \lambda\}$ for some $\lambda$. This is called the *density level-set* of $f$ at level $\lambda$. The cluster tree of $f$ is the hierarchy formed by the infinite collection of these clusters over all $\lambda$. Chaudhuri and Dasgupta [2] introduced Robust Single Linkage (RSL) which was the first cluster tree estimation procedure with precise statistical guarantees. Shortly after, Kpotufe and Luxburg [13] provided an estimator that ensured *false clusters* were removed using used an extra *pruning* step. Interestingly, Quick Shift does not require such a pruning step, since the points near cluster boundaries naturally get assigned to regions with higher density and thus no spurious clusters are formed near these boundaries. Sriperumbudur and Steinwart [14], Jiang [10], Wang et al. [17] showed that the popular DBSCAN algorithm [8] also estimates these level sets. Eldridge et al. [7] introduced the *merge distortion metric* for cluster tree estimates, which provided a stronger notion of consistency. We use their framework to analyze Quick Shift and show that this simple estimator is consistent in merge distortion.

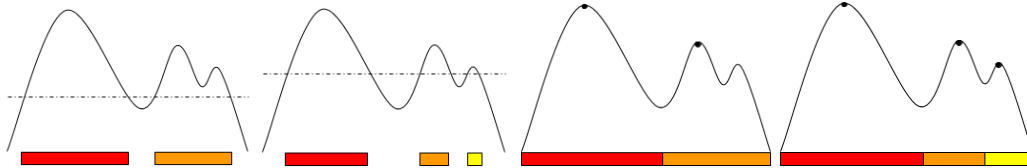

Figure 4: Density-based clusters discovered by level-set model $\{x : f(x) \geq \lambda\}$ (e.g. DBSCAN) vs Quick Shift on a one dimensional density. **Left** two images: level sets for two density level settings. Unassigned regions are noise and have no cluster assignment. **Right** two images: Quick Shift with two different $\tau$ settings. The latter is a hill-climbing based clustering assignment.

**Modal Regression.** Nonparametric modal regression [3] is an alternative to classical regression, where we are interested in estimating the modes of the conditional density $f(y|X = x)$ rather than the mean. Current approaches primarily use a modification of mean-shift; however analysis for mean-shift require higher order smoothness assumptions. Using Quick Shift instead for modal regression requires less regularity assumptions while having consistency guarantees.

# 8 Conclusion

We provided consistency guarantees for Quick Shift under mild assumptions. We showed that Quick Shift recovers the modes of a density from a finite sample with minimax optimal guarantees. The approach of this method is considerably different from known procedures that attain similar guarantees. Moreover, Quick Shift allows tuning of the segmentation and we provided an analysis of this behavior. We also showed that Quick Shift can be used as an alternative for estimating the

cluster tree which contrasts with current approaches which utilize proximity graph sweeps. We then constructed a procedure for modal regression using Quick Shift which attains strong statistical guarantees.

## Appendix

### Mode Estimation Proofs

**Lemma 4.** *Suppose Assumptions 1, 2, 3, 4, and 5 hold. Let $\bar{r} > 0$ and $h \equiv h(n)$ be chosen such that $h \to 0$ and $\log n/(nh^d) \to 0$ as $n \to \infty$. Then the following holds for $n$ sufficiently large with probability at least $1 - 1/n$. Define*

$$\tilde{r}^2 := \max\left\{\frac{32\hat{C}}{\check{C}}(\log n)^{4/\rho}h^2, 17 \cdot C'\sqrt{\frac{\log n}{n \cdot h^d}}\right\}.$$

*Suppose $x_0 \in \mathcal{M}$ and $x_0$ is the unique maximizer of $f$ on $B(x_0, \bar{r})$. Then letting $\hat{x} := \mathrm{argmax}_{x \in B(x_0,\bar{r}) \cap X_{[n]}} \widehat{f}_h(x)$, we have*

$$||x_0 - \hat{x}|| < \tilde{r}.$$

*Proof sketch.* This follows from modifying the proof of Theorem 3 of [11] by replacing $\mathbb{R}^d \backslash B(x_0, \tilde{r})$ with $B(x_0, \bar{r}) \backslash B(x_0, \tilde{r})$. This leads us to

$$\inf_{x \in B(x_0, r_n)} \widehat{f}_h(x) > \sup_{x \in B(x_0,\bar{r}) \backslash B(x_0,\tilde{r})} \widehat{f}_h(x),$$

where $r_n := \min_{x \in X_{[n]}} |x_0 - x|$ and $n$ is chosen sufficiently large such that $\tilde{r} < \tau$. Thus, $|x_0 - \hat{x}| \leq \tilde{r}$. $\qquad\square$

*Proof of Theorem 2.* Suppose that $x_0 \in \mathcal{M}^+_{\tau+\epsilon,\delta} \backslash \mathcal{M}^-_{\tau-\epsilon,\delta}$. Let $\hat{x} := \mathrm{argmax}_{x \in B(x_0,\tau) \cap X_{[n]}} \widehat{f}_h(x)$. We first show that $\hat{x} \in \widehat{\mathcal{M}}$.

By Lemma 4, we have $|x_0 - \hat{x}| \leq \tilde{r}$ where $\tilde{r}^2 := \max\left\{\frac{32\hat{C}}{\check{C}}(\log n)^{4/\rho}h^2, 17 \cdot C'\sqrt{\frac{\log n}{n \cdot h^d}}\right\}$. It remains to show that $\hat{x} = \mathrm{argmax}_{x \in B(\hat{x},\tau) \cap X_{[n]}} \widehat{f}_h(x)$. We have $B(\hat{x}, \tau) \subseteq B(x_0, \tau + \tilde{r})$. Choose $n$ sufficiently large such that (i) $\tilde{r} < \epsilon$, (ii) by Lemma 1, $\sup_{x \in \mathcal{X}} |\widehat{f}_h(x) - f(x)| < \delta/4$ and (iii) $\tilde{r}^2 < \delta/(4\hat{C})$. Now, we have

$$\sup_{x \in B(x_0, \tau+\tilde{r}) \backslash B(x_0,\tau)} \widehat{f}_h(x) \leq \sup_{x \in B(x_0, \tau+\tilde{r}) \backslash B(x_0,\tau)} f(x) + \delta/4 \leq f(x_0) - 3\delta/4$$

$$\leq f(\hat{x}) + \hat{C}\tilde{r}^2 - 3\delta/4 < f(\hat{x}) - \delta/2 < \widehat{f}_h(\hat{x}).$$

Thus, $\hat{x} = \mathrm{argmax}_{x \in B(\hat{x},\tau) \cap X_{[n]}} \widehat{f}_h(x)$. Hence, $\hat{x} \in \widehat{\mathcal{M}}$.

Next, we show that it is unique. To do this, suppose that $\hat{x}' \in \widehat{\mathcal{M}}$ such that $||\hat{x}' - x_0|| \leq \tau/2$. Then we have both $\hat{x} = \mathrm{argmax}_{x \in B(\hat{x},\tau) \cap X_{[n]}} \widehat{f}_h(x)$ and $\hat{x}' = \mathrm{argmax}_{x \in B(\hat{x}',\tau) \cap X_{[n]}} \widehat{f}_h(x)$. However, choosing $n$ sufficiently large such that $\tilde{r} < \tau/2$, we obtain $\hat{x} \in B(\hat{x}', \tau)$. This implies that $\hat{x} = \hat{x}'$, as desired.

We now show $|\widehat{\mathcal{M}}| \leq |\mathcal{M}| - |\mathcal{M}^-_{\tau-\epsilon,\delta}|$. Suppose that $\hat{x} \in \widehat{\mathcal{M}}$. Let $\tau_0 := \min\{\epsilon/3, \tau/3, r_M/2\}$. We show that $B(\hat{x}, \tau_0) \cap \mathcal{M} \neq \emptyset$. Suppose otherwise. Let $\lambda = f(\hat{x})$. By Assumptions 2 and 5, we have that there exists $\sigma > 0$ and $\eta > 0$ such that the following holds uniformly: $\mathrm{Vol}(B(\hat{x}, \tau_0) \cap L_f(\lambda + \sigma)) \geq \eta$. Choose $n$ sufficiently large such that (i) by Lemma 1, $\sup_{x \in \mathcal{X}} |\widehat{f}_h(x) - f(x)| < \min \sigma/2, \delta/4$ and (ii) there exists a sample $x \in B(\hat{x}, \epsilon/3) \cap L_f(\lambda + \sigma) \cap X_{[n]}$ by Lemma 7 of Chaudhuri and Dasgupta [2]. Then $\widehat{f}_h(x) > \lambda + \sigma/2 > \widehat{f}_h(\hat{x})$ but $x \in B(\hat{x}, \tau_0)$, a contradiction since $\hat{x}$ is the maximizer of the KDE of the samples in its $\tau$-radius neighborhood. Thus, $B(\hat{x}, \tau_0) \cap \mathcal{M} \neq \emptyset$. Now, suppose that there exists $x_0 \in B(\hat{x}, \tau_0) \cap \mathcal{M}^-_{\tau-\epsilon,\delta}$. Then, there exists $x' \in B(x_0, \tau - 2\tau_0)$

such that $f(x') \geq f(x_0) + \delta$. Then, if $\bar{x}$ is the closest sample point to $x'$, we have for $n$ sufficiently large, $|x' - \bar{x}| \leq \tau_0$ and $f(\bar{x}) \geq f(x_0) + \delta/2$ and thus $\widehat{f}_h(\bar{x}) > f(\bar{x}) - \delta/4 \geq f(\hat{x}) + \delta/4 > \widehat{f}_h(\hat{x})$. But $\bar{x} \in B(\hat{x}, \tau) \cap X_{[n]}$, contradicting the fact that $\hat{x}$ is the maximizer of the KDE over samples in its $\tau$-radius neighborhood. Thus, $B(\hat{x}, \tau_0) \cap (\mathcal{M} \backslash \mathcal{M}^-_{\tau-\epsilon,\delta}) \neq \emptyset$.

Finally, suppose that there exists $\hat{x}, \hat{x}' \in \widehat{\mathcal{M}}$ such that $x_0 \in \mathcal{M} \backslash \mathcal{M}^-_{\tau-\epsilon,\delta}$ and $x_0 \in B(\hat{x}, \tau_0) \cap B(\hat{x}', \tau_0)$. Then, $\hat{x}, \hat{x}' \in B(x_0, \tau_0)$, thus $|\hat{x} - \hat{x}'| \leq \tau$ and thus $\hat{x} = \hat{x}'$, as desired. $\qquad\square$

**Cluster Tree Estimation Proofs**

**Lemma 5** (Minimality)**.** *The following holds with probability at least $1 - 1/n$. If $A$ is a connected component of $\{x \in \mathcal{X} : f(x) \geq \lambda\}$, then $A \cap X_{[n]}$ is contained in the same component in $\widehat{C}_f(\lambda - \epsilon)$ for any $\epsilon > 0$ as $n \to \infty$.*

*Proof.* It suffices to show that for each $x \in A$, there exists $x' \in B(x, \tau/2) \cap X_{[n]}$ such that $\widehat{f}_h(x') > \lambda - \epsilon$. Given our choice of $\tau$, it follows by Lemma 7 of [2] that $B(x, \tau/2) \cap X_{[n]}$ is non-empty for $n$ sufficiently large. Let $x' \in B(x, \tau/2) \cap X_{[n]}$. Choose $n$ sufficiently large such that by Lemma 1, we have $\sup_{x \in \mathcal{X}} |\widehat{f}_h(x) - f(x)| < \epsilon/2$. We have $f(x') \geq \inf_{B(x, \tau/2)} f(x) \geq \lambda - C_\alpha(\tau/2)^\alpha > \lambda - \epsilon/2$, where the last inequality holds for $n$ sufficiently large so that $\tau$ is sufficiently small. Thus, we have $\widehat{f}_h(x') > \lambda - \epsilon$, as desired. $\qquad\square$

**Lemma 6** (Separation)**.** *Suppose that $A$ and $B$ are distinct connected components of $\{x \in \mathcal{X} : f(x) \geq \lambda\}$ which merge at $\{x \in \mathcal{X} : f(x) \geq \mu\}$. Then $A \cap X_{[n]}$ and $B \cap X_{[n]}$ are separated in $\widehat{C}_f(\mu + \epsilon)$ for any $\epsilon > 0$ as $n \to \infty$.*

*Proof.* It suffices to assume that $\lambda = \mu + \epsilon$. Let $A'$ and $B'$ be the connected components of $\{x \in \mathcal{X} : f(x) \geq \mu + \epsilon/2\}$ which contain $A$ and $B$ respectively. By the uniform continuity of $f$, there exists $\tilde{r} > 0$ such that $A + B(0, 3\tilde{r}) \subseteq A'$. We have $\sup_{x \in A' \backslash (A + B(0, \tilde{r}))} f(x) = \mu + \epsilon - \epsilon'$ for some $\epsilon' > 0$.

Choose $n$ sufficiently large such that by Lemma 1, we have $\sup_{x \in \mathcal{X}} |\widehat{f}_h(x) - f(x)| < \epsilon'/2$. Thus, $\sup_{x \in A' \backslash (A + B(0, \tilde{r}))} \widehat{f}_h(x) < \mu + \epsilon - \epsilon'/2$. Hence, points in $\widehat{C}_f(\mu + \epsilon)$ cannot belong to $A' \backslash (A + B(0, \tilde{r}))$. Since $A'$ also contains $A + B(0, 3\tilde{r})$, it means that there cannot be a path from $A$ to $B$ with points of empirical density at least $\mu + \epsilon$ with all edges of length less than $\tilde{r}$. The result follows by taking $n$ sufficiently large so that $\tau < \tilde{r}$, as desired. $\qquad\square$

*Proof of Theorem 4.* By the regularity assumptions on $f$ and Theorem 2 of [7], we have that Algorithm 2 has both uniform minimality and uniform separation (defined in [7]), which implies convergence in merge distortion. $\qquad\square$

**Modal Regression Proofs**

*Proof of Theorem 5.* There are two directions to show. (1) if $\hat{y} \in \widehat{M}_x$ then $\hat{y}$ is a consistent estimator of some mode $y_0 \in \mathcal{M}_x$. (2) For each mode, $y_0 \in \mathcal{M}$, there exists a unique $\hat{y} \in \widehat{\mathcal{M}}$ which estimates it.

We first show (1). We show that $[\hat{y} - \tau, \hat{y} + \tau] \cap \mathcal{M}_x \neq \emptyset$. Suppose otherwise. Let $\lambda = f(x, \hat{y})$. Choose $\sigma < \tau/4$. Then by Assumptions 2 and 5, there exists $\eta > 0$ such that taking $\epsilon = \tau/2$, we have that there exists $\delta > 0$ such that $\{(x, y') : y' \in [\hat{y} - \tau, \hat{y} + \tau]\} \cap L_f(\lambda + \delta)$ contains connected set $A$ where $\text{Vol}(A) > \eta$. Choose $n$ sufficiently large such that (i) there exists $y \in A \cap Y$, and (ii) by Lemma 1, $\sup_{(x', y')} |\widehat{f}_h(x', y') - f(x', y')| < \delta/2$. Then $\widehat{f}_h(x, y) > \lambda + \delta/2 > \widehat{f}_h(x, \hat{y})$ but $y \in [\hat{y} - \tau, \hat{y} + \tau]$, a contradiction since $\hat{y}$ is the maximizer of the KDE in $\tau$ radius neighborhood when restricted to $X = x$. Thus, there exists $y_0 \in \mathcal{M}_x$ such that $y_0 \in [\hat{y} - \tau, \hat{y} + \tau]$. Moreover this $y_0 \in \mathcal{M}_x$ must be unique by Lemma 2. As $n \to 0$, we have $\tau \to 0$ and thus consistency is established for $\hat{y}$ estimating $y_0$.

Now we show (2). Suppose that $y_0 \in \mathcal{M}_x$. From the above, for $n$ sufficiently large, the maximizer of the KDE in $[y_0 - 2\tau, y_0 + 2\tau] \cap Y$ is contained in $[y_0 - \tau, y_0 + \tau]$. Thus, there exists a root of the tree contained in $[y_0 - \tau, y_0 + \tau]$ and taking $\tau \to 0$ gives us the desired result. $\qquad\square$

## Acknowledgements

I thank the anonymous reviewers for their valuable feedback.

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
