[Reviews · NeurIPS 2017]

Reviewer 1



This paper provides theoretical guarantees for outputs of Quickshift-based algorithms, in three estimation problem: mode estimation, cluster tree estimation and modal regression. Up to my knowledge, such results for quickshift are new (though the same rates are obtained for mode estimation in ref 8 and RSL achieves consistency for the cluster tree estimation ref 3). Proofs are on the whole sound, but I have a few remarks: proof of Theorem 1, p.5, l.191: "by restricting f to B(hat(x),tau)". Since hat(x) depend on X_[n], this seems not that straightforward. You may easily get rid of this by proving (2) before (1). (2) yields that for every x0 there is a unique root in B(x0,2tau) and this root is the max of KDE over B(x0,2tau). So, coming back to (1), since you proved that hat(x) is tau close to some x0, then hat(x) must be the maximizer of kde over B(x0,2tau), the latter ball being not random and you can restric f. proof of Theorem 2, p.6, l.222: 'r < eps gives us hat(x) in hat(M)'. I do not see why: you have to prove that hat(x) is KDEmax over B(hat(x), tau) that is included in B(x0,tau + r). So you have to prove that hat(f)(hat(x)) > sup B(x0,tau + r)/B(x0,tau) hat(f). You can do that using the same argument as below, ll.232-236. p.6, Lemma 4: the n sufficiently large also depends on delta, no? p.7, Theorem 3: same remark as above concerning the dependency on delta. You should also mention the required hypothesis (Holder is enough? by the way I do not see where the assumption tau < r_M/2 matters) p.7, Theorem 5: the 'uniformly in x' might be misleading. For instance, conditional mode estimation involves some r_M(x), so n has to be large enough so that tau < = r_M(x). Some clarification bout this uniformity might be useful. Minor concerns p.2, l.58: about the work of Arias-Castro et al., bottom of p.9 it is proved that the uniform difference between the 'mean-shift' trajectory and the ideal one is decreasing (for 'a' small enough). Since the ideal trajectory converges eventually to a true mode, I guess that 'mean-shift' too. p.2, l.99: spurious '.' p.2, algorithm 1: 'X_i' - > 'x_i' p.4, Lemma 2: I guess A_x0 is the connected component that contains x0 P.4, l.162: 'folowing' p.5, l.192: spurious '.' p.5, l.200: 'intentially' is 'intentionally'? p.5, l.200: 'particlar' p.7, l.263: 'pairs vertices' p.7, l.280: missing 'we', and 'conditiona' p.8, l.298: 'there exist y0'

Reviewer 2



This paper provides finite sample bounds for Quick Shift in mode estimation, cluster recovery, and modal regression. It also shows that Quick Shift has the advantage of assigning sample points to their modes for clustering. These results are good statistical analysis for Quick Shift. However, there are two minor shortcomings in the presented analysis: 1. The decay assumption for the kernel function k stated as k(t) \leq C_\rho exp(-t^\rho) with \rho >0 is rather strong since polynomially decaying kernels are excluded. 2. In all the five theorems, the probability level is 1- 1/n which depends on the sample size and cannot be arbitrarily close to 1. The results in the paper are interesting in general.

Reviewer 3



In this paper the authors submit proofs regarding various notions of consistency for the Quick Shift algorithm. They also introduce adaptions of the Quick Shift algorithm for other tasks paired with relevant consistency results. Quick Shift is a mode finding algorithm where one chooses a segmentation parameter \tau, constructs a kernel density estimator, and assigns a mode, m, at samples where the KDE magnitude at m is larger than the KDE magnitude evaluated at all samples within a \tau-ball of m. In this setting a true mode are those points in the sample space where the density is maximized within a \tau ball. Quick shift can also be used to assign samples to a mode by ascending a sample to the nearest sample where the KDE magnitude is larger and repeating until a mode is reached. The authors present proofs of consistency for both of these tasks: mode recovery and mode assignment. The authors also propose an adaptation of Quick Shift for cluster tree recovery and modal regression for which they also prove consistency. The proofs presented in this paper are dense and technical; I cannot claim to have to have totally validated their correctness. That being said I found no glaring errors in the authors' reasoning. My largest concern with this paper is reference [1]. [1] plays a central role in the proofs for individual mode recovery rates. [1] is an "anonymous "unpublished manuscript" on Dropbox. I spent a fair amount of time searching for any other reference to this manuscript and I could find none. The authors give no explanation for this manuscript. I'm inclined to think that I cannot reasonably accept a paper whose results rely upon such a dubious source. Additionally there are many other smaller issues I found in the paper. Some of them are listed here by line: 99: period after "we" should be removed 101-103: The sentence "For example, if given an input X the response y have two polarizing tendancies, modal regression would be able to adapt to these tendancies," is very strange and unclear. Also "tendencies" is misspelled. 103: "indepth" is not a word. Replace with "in-depth" or "indepth" 106: "require" should be "requires" 112: Why is the "distribution \script{F}" introduced? This is never mentioned elsewhere. We can simply proceed from the density. 113: I'm guessing that the "uniform measure" is the same thing as the "Lebesgue measure." "Lebesgue measure" is more standard terminology in my opinion. 115: Why is the norm notation replaced with absolute value notation? This occurs later as well elements which are clearly vectors, line 137 and Lemma 2 for instance. 139: Remark 3 should have a citation. 145: C is never defined. 150: Your style changes here. Assumption 3 uses parentheses and here Assumption 4 uses square brackets. 158: The final "x" in the set definition should be "x'" 242: "f(x)" should be to the right of "inf" 280: "conditiona" should be "conditional" 286: What exactly is meant by "valid bandwidth choices?" Update: I've bumped the paper by two points. I still have concerns with the numerous small technical errors I found